# Use of Confectionery Waste in Biogas Production by the Anaerobic Digestion Process

**DOI:** 10.3390/molecules24010037

**Published:** 2018-12-21

**Authors:** Agnieszka A. Pilarska, Krzysztof Pilarski, Agnieszka Wolna-Maruwka, Piotr Boniecki, Maciej Zaborowicz

**Affiliations:** 1Institute of Food Technology of Plant Origin, Poznan University of Life Sciences, Wojska Polskiego 31, 60-637 Poznań, Poland; 2Institute of Biosystems Engineering, Poznan University of Life Sciences, Wojska Polskiego 50, 60-637 Poznań, Poland; pilarski@up.poznan.pl (K.P.); bonie@up.poznan.pl (P.B.); maciej.zaborowicz@up.poznan.pl (M.Z.); 3Department of General and Environmental Microbiology, Poznan University of Life Sciences, Wojska Polskiego 31, 60-637 Poznań, Poland; amaruwka@up.poznan.pl

**Keywords:** confectionery waste, anaerobic digestion, biodegradation, process stability, biogas and biomethane yields

## Abstract

It was the objective of this study to verify the efficiency and stability of anaerobic digestion (AD) for selected confectionery waste, including chocolate bars (CB), wafers (W), and filled wafers (FW), by inoculation with digested cattle slurry and maize silage pulp. Information in the literature on biogas yield for these materials and on their usefulness as substrate in biogas plants remains to be scarce. Owing to its chemical structure, including the significant content of carbon-rich carbohydrates and fat, the confectionery waste has a high biomethane potential. An analysis of the AD process indicates differences in the fluctuations of the pH values of three test samples. In comparison with W and FW, CB tended to show slightly more reduced pH values in the first step of the process; moreover an increase in the content of volatile fatty acids (VFA) was recorded. In the case of FW, the biogas production process showed the highest stability. Differences in the decomposition dynamics for the three types of test waste were accounted for by their different carbohydrate contents and also different biodegradabilities of specific compounds. The highest efficiency of the AD process was obtained for the filled wafers, where the biogas volumes, including methane, were 684.79 m^3^ Mg^−1^ VS and 506.32 m^3^ Mg^−1^ VS, respectively. A comparable volume of biogas (673.48 m^3^ Mg^−1^ VS) and a lower volume of methane (407.46 m^3^ Mg^−1^ VS) were obtained for chocolate bars. The lowest volumes among the three test material types, i.e., 496.78 m^3^ Mg^−1^ VS (biogas) and 317.42 m^3^ Mg^−1^ VS (methane), were obtained for wafers. This article also proposes a method of estimation of the biochemical methane potential (theoretical BMP) based on the chemical equations of degradation of sugar, fats, and proteins and known biochemical composition (expressed in grams).

## 1. Introduction

Due to the energy crisis and climatic changes the world is searching for an ecological and carbon-neutral source of energy, which could replace fossil fuels. The safety of supplying energy, especially renewable one, and the reduction of CO_2_ emission have become priorities. The microbiological process of anaerobic digestion (AD), which has been known for a long time, is a promising and cheap method of biogas production [1]. Organic waste, including food waste, is increasingly often used in an attempt to solve another problem of the civilised world, i.e., high production of waste [2,3,4]. This technology is both a recipe to minimise the harmful effect on the environment and it is a source of methane—the biofuel of the future.

Food waste is an easily biodegradable substrate [5,6]. The organic matter it contains is a valuable nourishment for bacteria. Food waste is characterised by high biomethane production potential (200–670 mL CH_4_ g^−1^ VS (volatile solids) added) [7,8]. According to reports in the literature, waste food from restaurants, individually or in combination with other cosubstrates, is usually subjected to the AD process [9]. There are also experiments on food waste from industrial production, such as: sugar beet pulp, molasses, cheese whey, fat, coffee waste, fruit and vegetable waste [10,11,12]. Experiments on confectionery waste are very rare [13,14], although it is usually a highly concentrated material rich in carbohydrates, which is a promising substrate for methane production.

The confectionery industry generates high amounts of confectionery waste in a continuous manner. It is a very important factor in view of potential biogas production investment projects. Tonnes of waste are produced in a typical enterprise every week; hundreds of tonnes are produced every year [15,16]. Solid waste is usually produced, whereas liquid waste is less frequent (usually post-process water). One of the most common types of solid waste is defective confectionery. Imperfectly shaped items, stuck together, broken or only defectively packed or incorrectly labelled, may reach up to 10% of the total confectionery production. Other kinds of waste produced in similar quantities are: dough, chocolate mass, fatty flavour fillings, starch from jelly production, etc. [14]. The confectionery industry mostly disposes of solid waste products through partial recycling and combustion. Waste utilisation through combustion has always been problematic, chiefly due to the high amounts of pollution emitted by waste combustion gases [16]. Confectionery waste is increasingly often recommended for the production of animal feed. However, it is necessary to consider the costs of initial processing, sterilisation, and supplementation. The direct application of food waste as animal feed involves the high risk of propagation of diseases as a result of a shorter food chain [5]. Among the methods listed, anaerobic digestion is the best alternative, as it is the most economical and friendly to the environment.

Even if anaerobic digestion of food waste is considered a proven technology, there are still some typical technical difficulties or problems related with the scientific understanding of the process specificity [17]. The pH value is one of the most important parameters, which is decisive to the course of organic matter decomposition, because it affects both chemical reactions and activity of the bacterial flora [9]. The optimal pH for the growth of methanogens is 6.5–7.2 [18]. A decrease in the pH value in the system (system, medium, environment—used interchangeably) below 6.5 is caused by the accumulation of volatile fatty acids (VFA), whose concentration is higher than the buffer capacity of the system [19,20]. The activity of VFA-decomposing (consuming) methanogens is often reduced, and in consequence, the production of biogas may be interrupted. When the pH value is higher than 7—there might be also negative consequences for the AD process [21]. Increased alkalinity affects the NH_3_ and NH_4_^+^ dissociation equilibrium. High pH and high temperature (in the thermophilic AD) favour the accumulation of NH_3_(aq), which is able to pass through microbial membranes, affecting the cellular osmoregulation and thus inhibiting the microbial performance.

The monitoring of the process stability should include not only pH measurements, but also measurements of VFA and/or total alkalinity (TA). On the one hand, the VFA behaviour provides information about the performance of the intermediate AD steps. On the other hand, alkalinity is the capacity of the digester medium (mixtures) to neutralise the VFA generated during the process and to affect pH changes. According to the literature data, the different ranges of the VFA/TA ratio are interpreted as follows: VFA/TA ≤ 0.40—stable digester, 0.40 < VFA/TA < 0.80—some signs of instability, and VFA/TA ≥ 0.80—significant instability [19].

The aim of most anaerobic digestion tests is to assess the biochemical methane potential (BMP). This parameter indicates the maximum methane potential of different organic substrates. BMP tests are a useful tool for determining the best substrate and codigestion configurations. However, there are some methods of prediction and/or verification of the final yield of methane based on the organic composition of substrates [22]. These methods save costs and time. Recently they have been presented and used by Nielfa et al. [23] and Zarkadas et al. [24]. Apart from that, Nielfa et al. [23], who studied the co-digestion of organic fraction of municipal solid waste and biological sludge, presented and applied another two methods of theoretical BMP estimation. One of them was based on the mass of the sample and the chemical oxygen demand (COD)concentration, and the other was based on the elemental composition (C, O, H, and N) of material in adequate equations. The same author used BMP mathematical models, which enabled reproduction of the methane curve behaviour and prediction of the final methane productions, beginning with the first days of experimentation. A full set of calculations based on different methodologies guarantees quick access to reliable information and indications for the best codigestion configuration.

The aim of present article was to analyse and compare biogas and biomethane yields of selected confectionery waste, including chocolate bars (CB), wafers (W), and filled wafers (FW) by inoculation with digested cattle slurry and maize silage pulp. The study was carried out on a laboratory scale in anaerobic batch reactors, at controlled (mesophilic) ranges of temperature, pH, and VFA/TA ratio.

## 2. Materials and Methods

### 2.1. Substrates and Inoculum

Confectionary waste (CW), including CB, W, and FW, was acquired from a manufacturer in Poznań, Poland. The inoculum in the form of digested cattle slurry and maize silage pulp was provided by a local agricultural biogas plant from Greater Poland Voivodeship. The inoculum was transported from the sewage disposal plant in a portable cooler with adjustable temperature and was used in the experiments without delay. Table 1 shows the physiochemical properties of the materials.

### 2.2. Experimental Setup

The first stage of the experiment consisted in the preparation of digestion mixtures in the form of three batches: CB/inoculum, W/inoculum and FW/inoculum (Table 2). The digestion mixture ratios were based on the Verein Deutscher Ingenieure (VDI) 4630 guideline [25], which concerns the digestion of organic materials, characterisation of substrates, sample taking, collection of material data and digestion tests. According to this guideline and literature data, the authors of the present study attempted to keep the total solids content (TS) in the batch below 10% to guarantee adequate mass transfers and the content of volatile solids (VS) between 1.5 and 2% in the batch with the inoculum. Before digestion the pH of the mixtures was characterised by a narrow range, i.e., 6.5–7.2.

Table 2 shows the compositions and selected parameters of the mixture.

The analyses of biogas production rates and biogas and methane yields were carried out according to the German standard DIN 38 414-S8 [26]. The AD process was carried out in a multichamber biofermenter (Figure 1). The pH values and variations in the VFA/TA ratio (Figure 2) as well as biogas production (Figure 3a,b) and biogas composition were monitored daily in each sample. Five millilitre samples of the digest mixture were drawn via slurry-sample drawing tube in the biofermenter (see Figure 1) under anaerobic conditions, by means of specially-selected syringes.

Twelve digestion chambers were used in the tests. Each substrate and the control sample (inoculum) were digested in triplicate. Adequate substrate mixtures were placed in 1.4 L biofermenters (5) with 1 L of the feed in each. The material was stirred every 24 h to prevent any uncontrollable decay of the organic matter. Each biofermenter was equipped with a water jacket (4) connected to a heater (1). This enabled control of the temperature and performance of the process at desirable temperatures. The tests were carried out under mesophilic conditions (at approx. 39 °C). The resulting biogas was transported through a tube into tanks (7) filled with a neutral liquid (8). In accordance with the VDI 4630 guideline, the experiment was conducted for each substrate until the daily biogas production was lower than 1% of the total amount generated [15,25,27].

### 2.3. Analytical Methods

Before and during fermentation the substrates, inoculum and batch were analysed according to applicable standards/procedures shown in Table 3.

The generated gas volumes were measured every 24 h. A qualitative analysis of the gas was carried out for the gas volumes not lower than 1 L, initially once a day, then—as lower volumes were generated—every three days. The concentrations of methane, carbon dioxide, hydrogen sulphide, ammonia, and oxygen were measured using a Geotech GA5000 gas analyser (Tusnovics Instruments, Kraków, Poland). The gas analyser measures gas concentrations in the following ranges: 0–100% CH_4_, 0–100% CO_2_, 0–25% O_2_, 0–2000 ppm H_2_S, and 0–1000 ppm NH_3_, respectively. The gas monitoring system was calibrated once a week by means of calibrating mixtures from Air Products and Chemicals Inc (Alletown, Pennsylvania, USA). The calibrating gas mixtures were used at the following concentrations: 65% CH_4_, 35% CO_2_ (in a single mixture), as well as 500 ppm H_2_S and 100 ppm NH_3_. Synthetic air, with an O_2_ content of 20%, was used for calibration of O_2_.

### 2.4. Calculation of Cumulative Biogas and Methane

After the qualitative and quantitative analyses of the gas obtained, the final step is to assess the biogas yield per unit of organic dry matter (m^3^ Mg^−1^). The calculations are based on the test results at standard temperature and pressure (STP). The biogas yield for the substrates is calculated by subtracting the gas volume generated for the inoculum. For the batches in the reactors filled with the substrate mixtures, the ratio of gas generated from the inoculum in the test is calculated from the following equation:(1)$$V_{IS(corr.)}=\frac{\sum V_{IS}m_{IS}}{m_{M}}$$
where V_IS(corr.)_ is the gas volume, released from the inoculum (mL_N_); ΣV_IS_ is the total gas volumes in the test performed on inoculum for the given test duration (mL_N_); m_IS_ is the mass of the inoculum used for the mixture (g); and m_M_ is the mass of the inoculum used in the control test (g).

Statistical analyses (standard deviation) were conducted by means of Statistica 12.0 software (Publisher, City, US State if US). We used one-way ANOVA analysis of variance to determine the significance of variation in the chemical analysis of the substrates and inoculum. Moreover, Least Significant Difference (LSD) tests were also used and their results are presented in order to facilitate an interpretation of the obtained differences at the level of the parameters under study. Pearson’s linear correlation coefficient was used to determine the correlation between methane yields and the chemical composition of the substrates.

## 3. Results and Discussion

### 3.1. Characterisation of Substrates and Biodegradation

The one-way ANOVA analysis of variance and the least significant difference test have been performed, which indicate that considerable differences in the chemical compositions exist between the experimental objects under study (see Table 1).

The confectionery waste (CW) used in present experiment was characterised by neutral pH values (6.62–7.84). It was the most favourable pH range for anaerobic digestion of FW (pH = 7.02) (see Table 1). The materials used in the experiment were characterised by lower conductivities than other kinds of food waste (except the inoculum), ranging from 1.21 to 1.85 mS cm^−1^ [32,33]. This indicates that they had a low content of dissolved minerals. The conductivity values noted in the experiment were confirmed by the results obtained for light metal ions (Table 1). The contents of K, Na, Mg, and Ca were too low to inhibit the process (for instance, by precipitation of carbonate and phosphate or undesirable neutralisation of the membrane potential) [19]. The types of CW used in this experiment had comparably high contents of total solids (TS) and volatile solids (VS). Their high values of total organic carbon (TOC) resulted from the predominance of carbohydrates. The highest TOC/TKN ratio (TKN—total Kjeldahl nitrogen) corresponded to that of chocolate bars (52.6), which resulted from the high share of carbohydrates (489.8 g kg^−1^ sucrose and 651.9 g kg^−1^ starch) and the low share of crude proteins (27.6 g kg^−1^). However, the value was far from the optimum TOC/TKN ratio, which, according to literature reports, improves the functioning of methanogens within the values ranging from 25 to 30 [2]. The other two materials, except the inoculum, had more favourable values of the parameter.

Since carbohydrates, mostly starch (Table 1), were the predominant component of the substrates used in present experiment, the first stage of biodegradation (hydrolysis) can generally be described with the following equation:(2)$${\left(C_{6}H_{10}O_{5}\right)}_{n}+nH_{2}O\to nC_{6}H_{12}O_{6}$$

Water breaks α-glycoside bonds between glucose mers—the structural material of starch. Hydrolysis (Equation (2)) is catalysed by extracellular microbial enzymes known as hydrolyses or lyses [2]. Although a qualitative analysis of biodegradation of the confectionery waste tested has not been provided in the presented study, it is worth noting that the subsequent steps of decomposition of organic matter (also CW) are accompanied by the release of VFA, alcohols, and aldehydes [32], which are finally converted into CO_2_ and CH_4_ by acetoclastic methanogens.

The biogas and methane yield of the CW used in the investigations was influenced not only by carbohydrates but also by fats, proteins and fibre (Table 1). Deublein and Steinhauser [34] in their book presented general equations describing the degradation of various organic materials into biogas, distinguishing between carbohydrates, proteins and fats.
(3)$$\mathrm{Carbohydrates}:\;C_{6}H_{12}O_{6}\to 3CO_{2}+3CH_{4}$$
(4)$$\mathrm{Fats}:\;C_{12}H_{24}O_{6}+3H_{2}O\to 4.5CO_{2}+7.5CH_{4}$$
(5)$$\mathrm{Proteins}:\;C_{13}H_{25}O_{7}N_{3}S+6H_{2}O\to 6.5CO_{2}+6.5CH_{4}+3NH_{3}+H_{2}S$$

These biochemical reactions are important for studies on the methane yield. They enable estimation of biochemical methane potential (BMP), which can be obtained by degradation of a given organic material if its biochemical composition is known. The calculated methane production (based on the molecular weights and weights obtained from the analyses of sugar, fat, and protein; under normal conditions), expressed in m^3^, as described in Section 3.3, can be converted into the amount of energy [35,36]. Like the earlier method developed and used by researchers, the method proposed in present study is based on similar data, i.e., the organic composition of the substrate. According to reports in the literature, the theoretical BMP can also be calculated if we know the percentage of protein, fat and carbohydrate fractions of VS and apply the adequate conversion formula [23,24,37].

As results from the calculations based on Equations (3)–(5) and the biochemical composition of the substrates used in present study (Table 1), the theoretical maximum amounts of methane which could be obtained from CB and FW were comparable, and they were greater than the amounts that could be obtained from W (as was mentioned). However, the actual yield of methane from chocolate bars—obtained in the experiment—was lower than the estimated theoretical value. It is most likely that this situation was caused by problems encountered during the process, which are described in Section 3.2 and Section 3.3.

### 3.2. Process Stability

The analysis of pH and VFA/TA ratio curves corresponding to the substrates used in the experiment (see Figure 2) showed that the anaerobic digestion of FW was the most stable process. The pH dropped only slightly to about 6.6 in the first phase of the AD process (until day 5), whereas the VFA/TA ratio increased to 0.45. At the consecutive stages of the process the pH of the digestion mixture gradually increased to 7.5, whereas the VFA/TA ratio dropped to 0.26 due to the degradation (depletion) of organic matter. However, the changes in the two parameters were within tolerable limits for methanogens [18] and did not affect the bacterial activity or biogas production, including methane. This observation was confirmed by the results of biogas production and the shape of the curves for filled wafers, shown in Figure 3a,b. The retention time for FW was 37 days, shorter than the retention time noted for the other substrates.

The situation was much less favourable for the other two substrates—CB and W, because the process of anaerobic digestion was much less stable. The curves shown in Figure 2 indicate a rapid decrease in the pH to 6.20 (W) and 6.28 (CB) on day 9. It was caused by the accumulation of VFA, as the VFA/TA ratio increased to 0.57 (W) and 0.63 (CB). However, it is noteworthy that the accumulation of VFA and reduction of pH during the initial phases of anaerobic digestion is common, especially in batch systems, due to methanogens’ lag and slow response to the consumption of the substrates provided to them by acidogens [24]. Further analysis of the diagram in Figure 2 shows that between days 15 and 21, the pH of the wafer digestion mixture decreased again but at a muchslower rate. During the consecutive days of the AD process the pH of both substrates gradually increased, whereas the VFA/TA ratio decreased. The biodegradation of W was slower than that of FW, as it lasted 43 days. However, the longest retention time, i.e., 59 days, was noted for CB.

The higher initial dynamics of the AD process in W and CB may have been caused by their chemical composition (Table 1). Unfilled W had a rather low content of sucrose (11.2 g kg^−1^) but higher content of starch than FW (757.7 g kg^−1^). There was also a relatively high content of starch in CB (651.9 g kg^−1^). As was mentioned earlier, starch is a polysaccharide composed only of glucose mers connected with α-glycoside bonds [38]. It is likely that easy breaking of these bonds caused premature onset of the acidogenic phase and accumulation of VFA. The biogas yield from chocolate bars was lower than expected. Their degradation took nearly twice as long as the degradation of FW and the cumulative production at the initial phases of the AD process was noticeably reduced, as can be seen in Figure 3a,b.

On the other hand, the statistical data indicate that, regardless of the substrate type, BMP correlated positively only with the protein content; for the chocolate bars and filled wafers, the correlation was statistically significant (see Table 4). Sucrose correlated positively with the yields only for the wafers. Moreover, no statistically significant correlation was found between the content of fat and starch vs. BMP in either of the experimental variants used. The fibre content showed some statistical significant correlation with BMP in the chocolate bars. Fiber is not a readily biodegradable carbohydrate because it consists mainly of cellulose and hemicellulose which are hardly water-soluble compounds; the fact could also have contributed to the lower biogas yield [36].

There are very few reports on the anaerobic digestion of CW. Lafitte-Trouqué and Forster [13] tested three configurations for a dual digestion system, using a mixture of sewage sludge and confectionery waste. The most serious problems were caused by the digester operating under thermophilic conditions (55 °C) due to very strong acidification resulting from the release of VFA (pH 3–4). However, the highest efficiency was observed in the configuration with the first stage operating at 55 °C and a secondary digester at 35 °C. This configuration also maintained a more stable pH. Recently the results of research on the semicontinuous mesophilic AD of waste wafer materials for a batch of 500 kg were published by Rusín et al. [14], who investigated the process stability by measuring the corresponding parameters: pH the VFA/TA ratio. During the first phase the pH of the wafers/inoculum digestion mixture was about 8. While the pH was decreasing to 6.8–7.2 during the AD process, the FOS/TAC was increasing to as high as 2–3. During the stabilisation phase the FOS/TAC ranged from 0.3 to 0.4. However, our experiment cannot be compared with the experiment conducted by Rusín et al. [14] because the author carried out a semi-continuous anaerobic digestion at high loads. In contrast, the authors of the presented research work carried out their experiment in a batch reactor, at reduced reactor loads, according to the standard VDI 4630 [25].

### 3.3. Biogas Production

The analysis of the biogas and methane yields with reference to fresh matter (FM) and volatile solids (VS) showed the highest yields from FW (Table 5). The yields of biogas and methane from chocolate bars CB were slightly lower, whereas W gave the lowest yields due to their biochemical composition. They contained much less fat and polysaccharides (sucrose) than the other substrates. The following amounts of methane were obtained from the fresh matter of FW, CB and W: 483.35 m^3^ Mg^−1^, 379.74 m^3^ Mg^−1^ and 306.55 m^3^ Mg^−1^, respectively. In terms of volatile solids were produced: 506.32 m^3^ Mg^−1^, 407.46 m^3^ Mg^−1^ and 317.42 m^3^ Mg^−1^, respectively.

As was mentioned, the theoretical yields of methane from FW and CB were comparable. The formulas of biochemical reactions 3–5 were used to calculate methane production from the substrates. The following values were obtained: 572.10 m^3^ Mg^−1^ VS, 349.14 m^3^ Mg^−1^ VS, and 580.55 m^3^ Mg^−1^ VS for CB, W, and FW, respectively (Table 4). The theoretical yield from FW was only slightly higher than the actual yield obtained in the experiment, which proves that the AD process ran correctly (see Table 5). The yield of methane from CB noted in the experiment was lower than the theoretical value. As can be seen in Figure 2, this situation was caused by destabilisation (acidification) of the system. As far as the yields from Ware concerned, the situation was similar to the yields from FW. The calculated methane yield was in agreement with the actual yield obtained in the experiment.

The reason why so different yields of methane were obtained for the confectionery waste tested was also their chemical compositions were different (Table 1). Their different levels of the various components—having different biodegradabilities—could have resulted in their methane yields being lower than the theoretical BMP. The probability that relationship has occurred can be related to the fiber, showing a statistically significant relationship with yields for chocolate bars, sucrose, and positively correlating with the biogas volume generated from wafers.

The biogas and biomethane yields from the CW used as individual substrates in this experiment were comparable even to the yield from fat [39,40] and higher than the yield from selected food waste, such as molasses, whey, fruit and vegetables [41,42,43,44]. The biogas obtained from the materials had a particularly high content of methane (60.5–64.5%)—higher than from other types of waste. The result was comparable to the results noted by other authors in AD experiments on wafers [13,16].

## 4. Conclusions

The results of the batch mesophilic AD of confectionery waste showed that the materials aresuitable and promising for biogas production. Substantially, this type of waste has never been studied or used as a substrate in biogas plants but the test results explicitly indicate that there it is realistic, potential competition to other currently-used materials. The high energy potential of the confectionery waste resulted from its high content of total solids and biochemical composition.

The cumulative biogas and methane production in terms of fresh and volatile solids were comparable for all the test substrates—this is their advantage because of costs of transport. As far as volatile solids are concerned, the following amounts of methane were obtained: from filled wafers—506.32 m^3^ Mg^−1^ VS, from chocolate bars—407.46 m^3^ Mg^−1^ VS and from wafers—317.42 m^3^ Mg^−1^ VS. The biogas produced from the substrates had a very high content of methane (up to 73.9% from filled wafers). The theoretical BMP calculated on the basis of the reactions of carbohydrates, fats, and proteins biodegradation indicate that it is possible to obtain a higher yield of methane from chocolate bars (572.10 m^3^ Mg^−1^ VS) than the yield obtained in present study. The results of analyses indicate that the methane yield from this substrate was reduced due to the adverse course of the AD process. Acidification of the environment, resulting in the process destabilization, have contributed to the fact that the results were obtained lower than expected. This could have also been caused by the positive correlation of chocolate bars with the slowly decomposable fibre. As far as filled wafers and wafers are concerned, the theoretical methane yields were very similar to the actual yields noted in the experiment (580.55 m^3^ Mg^−1^ VS; 349.14 m^3^ Mg^−1^ VS, respectively).

Other, advanced studies on the confectionery waste are envisaged, especially on filled wafers which have provided the highest yield of biogas (and methane) in this study. Potential cosubstrates for the confectionery waste will have to be investigated in our further studies. It will also be necessary to perform some biochemical and microbiological analyses when running the process. Moreover, it is expected that the use as inoculum of a stabilized sewage sludge with a considerable buffer capacity will provide an efficient solution to the problems (such as low pH in the digested medium) observed in the first digestion step. Moreover, it is envisaged that natural microbiological substrates will be used as additives in the anaerobic digestion confectionery waste to increase the process efficiency for every product (wafers, chocolate bars, and other ones), by improving the condition and stability of bacterial flora [45].

## Figures and Tables

**Figure 1 molecules-24-00037-f001:**
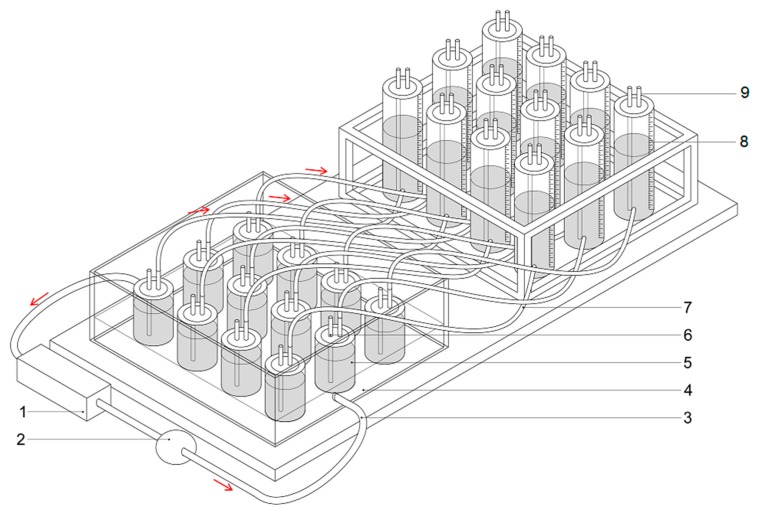
Biofermenter for biogas production tests (18-chamber section): 1—water heater with temperature adjustment; 2—water pump; 3—insulated tubes for liquid heating medium; 4—water jacket (39 °C); 5—biofermenter (1.4 dm^3^); 6—slurry-sample drawing tube; 7—tube for transporting the biogas; 8—graduated tank for biogas; 9—gas sampling valve.

**Figure 2 molecules-24-00037-f002:**
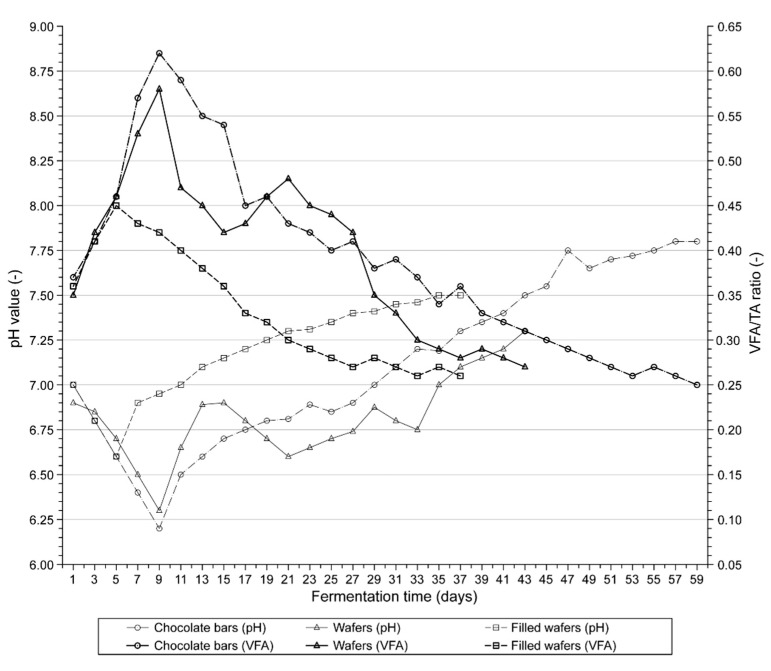
Variation in pH and VFA/TA during anaerobic digestion process of CB, W, and FW.

**Figure 3 molecules-24-00037-f003:**
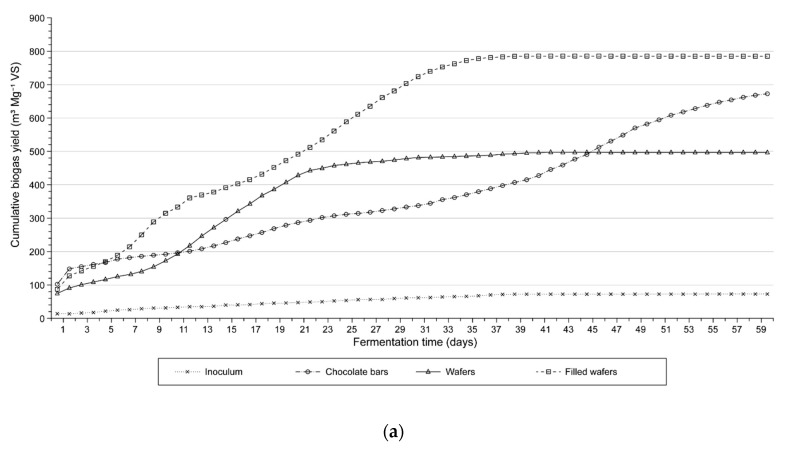
Cumulative yield of: (**a**) biogas and (**b**) methane from VS of inoculum, CB, W, and FW.

**Table 1 molecules-24-00037-t001:** Characteristics of substrates and inoculum (mean values, with standard deviation values in brackets).

Indicator	Unit	Chocolate Bars (CB)	Wafers (W)	Filled Wafers (FW)	Inoculum	LSD_0.05_
General						
pH	–	6.62 (0.10)	7.84 (0.33)	7.02 (0.27)	7.68 (0.04)	0.46
Cond.	mS cm^−1^	1.21 (0.04)	1.76 (0.03)	1.85 (0.06)	26.50 (0.62)	0.66
TS	wt %	94.80 (0.81)	97.69 (0.28)	96.77 (0.05)	3.18 (0.02)	0.31
VS	wt %_TS_	98.31 (1.02)	98.44 (0.05)	98.86 (0.03)	70.43 (0.22)	1.10
TOC	wt %_TS_	45.2 (0.53)	41.6 (0.36)	43.9 (0.40)	32.2 (0.36)	0.87
TKN	wt %_TS_	0.86 (0.01)	1.25 (0.02)	0.98 (0.03)	2.91 (0.02)	0.048
TOC/TKN ratio	–	52.6 (0.65)	33.3 (0.70)	44.8 (0.26)	11.0 (0.55)	1.21
TKN ^a^	mg kg^−1^	4.41 (0.04)	13.78 (0.03)	9.48 (0.04)	ND ^b^	3.21
TAN	wt %_TS_	0.26 (0.01)	0.23 (0.04)	0.33 (0.02)	2.6 (0.10)	0.12
P_total_	wt %_TS_	0.53 (0.04)	0.45 (0.04)	0.62 (0.03)	0.30 (0.03)	0.07
COD	mg L^−1^	1875 (6.55)	1128 (2.64)	1410 (4.58)	1643 (3.60)	87.11
Light metal ions						
K	mg kg^−1^	74.5 (0.43)	55.3 (0.17)	51.5 (0.36)	65.1 (0.43)	5.77
Na	mg kg^−1^	151.1 (0.85)	135.3 (0.26)	168.2 (0.30)	35.9 (0.17)	11.01
Mg	mg kg^−1^	35.0 (0.45)	41.1 (0.91)	32.3 (0.61)	10.3 (0.36)	1.31
Ca	mg kg^−1^	71.2 (2.16)	56.5 (1.86)	67.0 (1.96)	30.8 (0.36)	5.66
Biochemical composition						
Crude protein ^a^	g kg^−1^	27.6 (0.45)	84.3 (0.36)	59.3 (0.36)	ND	35.27
Crude fat	g kg^−1^	212.2 (0.72)	45.0 (0.36)	282.5 (0.72)	ND	46.78
Crude fiber	g kg^−1^	14.5 (0.26)	51.9 (0.52)	32.0 (0.55)	ND	15.97
Carbohydrate						
Sucrose	g kg^−1^	489.8 (1.60)	11.2 (0.17)	419.4 (0.65)	ND	45.21
Starch	g kg^−1^	651.9 (0.17)	757.7 (0.85)	591.1 (1.01)	ND	11.47

^a^ Protein: TKN × CF; CF: 6.25 for crude protein; ^b^ ND: not determined; LSD: Least Significant Difference. Cond.: conductivity, TS: total solids, VS: volatile solids, TOC: total organic carbon, TKN: total Kjeldahl nitrogen, TAN: total ammonium nitrogen, P_total_: total phosphorus, COD: chemical oxygen demand.

**Table 2 molecules-24-00037-t002:** Batch characteristics (mean values, with standard deviation values in brackets).

Batch	Substrate (g)	Inoculum (g)	pH	TOC/TKN Ratio	TS (%)
CB/inoculum	50	1000	6.82 (0.02)	15 (0.70)	7.54 (0.10)
W/inoculum	50	1000	6.94 (0.05)	17 (0.26)	7.68 (0.10)
FW/inoculum	50	1000	7.01 (0.11)	16 (0.26)	7.63 (0.05)

**Table 3 molecules-24-00037-t003:** Analytical methods.

Parameter	Method and Standard
pH	Potentiometric analysis (Elmetron CP-215, Elmetron, Zabrze, Poland); PN-EN 12176:2004, EN 15933:2012
TS	Gravimetric analysis, 105 °C (dryer Zalmed SML 30, Zalmed, Łomianki, Poland); PN-EN 12880:2004, EN 15934:2012
VS	Gravimetric analysis, 550 °C (furnace MS Spectrum PAF 110/6, Protherm Furnaces, Ankara, Turkey); PN-EN 12879:2004, EN 15935:2012
Cond.	Conductivity analysis (Elmetron CP-215, Elmetron, Zabrze, Poland); PN-EN 27888:1999.
TOC	Combustion (900 °C), CO_2_ determination (Infrared Spectrometry, O-I analytical analyser, SRA Instruments, Lyon, France); PB/PFO-37, EN 15936:2012
TKN	Titration, Kjeldahl method, 0.1n HCl, Tashiro’s indicator; PN-EN 13342, EN 15104:2011
TAN	Distillation and titration an method, 0.1n HCl, Tashiro’s indicator; PN-ISO 5664, ISO 5664
P_total_	Mineralization of phosphorus compounds with nitric acid (microwave furnace, Milestone, Hanon Instruments, Jinan, China), spectrophotometric analysis (Varian Cary 50, Varian Medical System, Palo Alto, CA, USA); PB/PFO-11, EN 14672:2005
VFA/TA ratio *	Titration with 0.05 M H_2_SO_4_ to two end values (pH 5.0 and 4.4)
COD	Titration, dichromate method (potassium dichromate, concentrated sulphuric acid, silver sulfate as catalyst); PN-ISO 6060-2006
Light metal ions	Inductively coupled plasma optical emission spectrometry (ICP-OES, JY 2000 2 ICP-OES Spectrometer, Hitachi, Tokyo, Japan); PN-EN ISO 11885:2009
Crude proteins	Calculated from TKN using a conversion factor of 6.25 for crude proteins; AOAC 920.87 [28]
Crude fats	Soxhlet method; extracted with hexane by using an automatic extractor Soxhlet model B-811 BUCHI, (Büchi Labortechnik AG, Flawil, Switzerland); AOAC 920.85 [29].
Crude fibre	Chemical method (digestion in 0.25N H_2_SO_4_ and then 0.25N NaOH) AOAC 962.09 [30]
Carbohydrates	Phenol–sulphuric acid methods [31]

* VFA/TA ratio: volatile fatty acids/alkalinity ration.

**Table 4 molecules-24-00037-t004:** Pearson’s linear correlation coefficient between methane yields and chemical composition of substrates.

Substrates	Proteins	Fats	Fiber	Sucrose	Starch
Chocolate bars	0.97 *	0.80	0.98 *	−0.72	−0.17
Wafers	0.78	−0.78	0.72	0.96 *	0.79
Filled wafers	0.97 *	0.56	−0.63	−0.75	0.85

* correlation coefficient significant at significance level *p* > 0.05.

**Table 5 molecules-24-00037-t005:** Cumulative biogas and methane yields.

Batch	Biogas		Methane		CH_4_	Theoretical BMP
	(m^3^ Mg^−1^ FM)	(m^3^ Mg^−1^ VS)	(m^3^ Mg^−1^ FM)	(m^3^ Mg^−1^ VS)	(%)	(m^3^ Mg^−1^ VS)
Inoculum	1.61 (0.16)	71.91 (7.12)	0.51 (0.07)	22.94 (5.17)	31.9 (6.69)	–
CB/inoculum	627.67 (5.37)	673.48 (19.97)	379.74 (10.13)	407.46 (5.90)	60.5 (1.66)	572.10
W/inoculum	479.77 (27.13)	496.78 (57.85)	306.55 (13.71)	317.42 (8.88)	63.9 (5.85)	349.14
FW/inoculum	749.19 (9.61)	684.79 (18.59)	483.35 (12.92)	506.32 (5.32)	73.9 (8.72)	580.55

* FM: fresh matter, VS: volatile solids, BMP: Biochemical Methane Potential.

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
