# Peer review of "Use of Confectionery Waste in Biogas Production by the Anaerobic Digestion Process"

_molecules, 2018, doi:10.3390/molecules24010037_

Round 1
Reviewer 1 Report
The novelty of the manuscript is low. There are many scientific publications dealing with BMP, comparing theretical and empirical data in literature. In fact, I would recomemend to updated the references.
Although the authors performed BMP following a well stablished method, they applied a really basic theortical calculation when more accute and also related to anaerobic digestion dynamics are available in literature. This manuscript is more suitable for academic purposes than for scientific dissemination.
Author Response
Poznań, 07-12-2018
Dear Reviewer,
I wish to thank you for your time, for reading and evaluating the paper written by myself and my colleagues, and for your suggestions.
- Several latest references have been added, as suggested.
Best regards,
Agnieszka Pilarska

Reviewer 2 Report
The article entitled "Use of confectionery waste in biogas production by anaerobic digestion process" should be accepted with minor revisions.
The manuscript is interesting and the topics well described the suitability of using confectionery waste for biogas production. The manuscript is also well written and the authors justify clearly the study, supported by a good literature review. Methods follow the state of the art and the results are reported accordingly.
The only issue is related to the significance of the results obtained because of the poor statistical analyses. For instance, it might be interesting to report some correlation between the BMP values and the composition of the materials used.
This and the other considerations are elaborated more thoroughly in the detailed response below:
Introduction:
The introduction is well written and the study is justified.
Line 55. Please introduce the abbreviation meaning.
Line 97. The authors report that VFA/TA ratio > 0.8 indicated a significant stability. Please check the reference literature, since this ratio means significant instability.
Materials and methods:
Table 1. I suggest to report in the Table caption if the values in the brackets are the SE or SD.
Line 186 “Calculation of cumulative biogas and methane”. It is not clearly reported if the volume of biogas was measured at STP.
Results and discussion:
Line 215. In my opinion the authors can use directly the abbreviations throughout the text.
Lines 232-238. This paragraph has to be rephrased in relation to the study obtained because it is a general description of the AD process.
Lines 300-311. It could be interesting to report the correlation between biogas yield and the composition of the materials used.
Lines 324-326. Please explain the differences with the experiment conducted by Rusin et al., otherwise this sentence is not useful.
Figure 2 and 3. Please report the SD in both figures.
Conclusions:
It is better to focus on the message of the study rather than on the main results obtained. Please, summarize and rephrase the conclusions.
Author Response
Poznan, 07-12-2018
Dear Reviewer,
I wish to thank you for your time, for reading and evaluating the paper written by myself and my colleagues. Thank you very much also - for your good opinion on my paper and for your valuable suggestions.
Statistical information has been added which explains the correlations between the methane yield of the test substrates and their biochemical composition.
We used one-way ANOVA analysis of variance to determine the significance of variation in the chemical analysis of the substrates and inoculum. Moreover, Least Significant Difference (LSD) tests were also used and their results are presented in order to facilitate an interpretation of the obtained differences at the level of the parameters under study. Pearson's linear correlation coefficient was used to determine the correlation between methane yields and the chemical composition of the substrates.
Introduction:
- The meaning of the abbreviation has been explained.
- The error has been corrected, the remark was justified;
Materials and methods:
- Additional explanations have been provided in the Table 1, 2, 3 captions (the values in the brackets are SD).
- This inconsistency has been explained.
Results and discussion:
- Abbreviations (especially for substrates) have been introduced throughout the text.
- This paragraph has been reworded, as suggested.
- Differences with the experiment conducted by Rusin et al. have been explained.
- At that point, a statistic report of the correlation between the biogas yield and the composition of the materials used has been introduced.
- Several summarizing conclusions have been added and plans for future studies on confectionery waste have been defined. The conclusions have been substantially reworded in response to the suggestions of all the Reviewers (there were different suggestions).
Best regards,
Agnieszka Pilarska

Reviewer 3 Report
The paper has elements of novelty, since the specific substrates described have not been studied extensively in the literature. However, my main concern is the lack of statistical analysis. Although triplicates were performed per treatment type, I see no ANOVA to check statistical differences among treatments. This needs to be added since it will clarify some of the conclusions and strengthen the work.
In addition, the authors present a lot of theoretical equations in their results and discussion. These equations belong to the background and should be part of the introduction or at least materials and methods.
The sampling procedure of the specific subtrates is not well described. Just two lines are provided (122-123). Give details. Amounts, how was the sampling performed, how were they transferred and stored each. The same comments applies to the inoculum.
The authors need to show the ISR achieved in the mixtures since this is the classical criterion to inoculate subtrates during anaerobic digestion. The basis on which the amounts of inoculum were added is not clear.
The authors need to present in a Table the comparision between the theoretical yields (according to their equations) and the experimental yields. What are the differences and what does this mean in terms of the biodegradable fraction of those wastes?
I see no Table with the initial composition of all substrates in terms of carbhydrates, proteins, fats, etc. Where is this information which is necessary in order to apply equations 3 to 5?
Conclusions 2 and 3 should be become quantitative. Explain in the conclusions the differences between experimental and theoretical values and present the biodegradable fractions (that should have been discussed earlier in the R&D section).
Show some error bars in 2 and 3 to reveal the variance among replications.
Author Response
Poznan, 07-12-2018
Dear Reviewer,
I wish to thank you for your time, for reading and evaluating the paper written by myself and my colleagues. Thank you for your valuable suggestions and for indicating the source of more information. It will certainly be of use in my future work.
- Statistical information has been added which explains the correlations between the methane yield of the test substrates and their biochemical composition.
We used one-way ANOVA analysis of variance to determine the significance of variation in the chemical analysis of the substrates and inoculum. Moreover, Least Significant Difference (LSD) tests were also used and their results are presented in order to facilitate an interpretation of the obtained differences at the level of the parameters under study. Pearson's linear correlation coefficient was used to determine the correlation between methane yields and the chemical composition of the substrates.
- The description of the sampling procedure has been elaborated (sampling during the analyses, and transport of the inoculum); The inoculum was transported from the sewage disposal plant in a portable cooler with adjustable temperature and was used in the experiment without delay.
- Guidelines concerning determination of the amount of inoculum have been provided in Methods ‘The author based her study on this guideline and literature data and attempted to keep the total solids content (TS) in the batch below 10% to guarantee adequate mass transfers and the content of volatile solids (VS) between 1.5 and 2% in the batch with the inoculum’ [VDI 4630 guideline]
The substrate/inoculum ratios in the batch are given in Table 2 (50 g substrate/1000 g inoculum)
- Theoretical values have been added in Table 5. The additional explanations resulting from the statistics (see table 4), are presented below table 5.
- The problem of the destabilization of the process has been discussed in section 3.2.
- I would appreciate it if you had a look – Table 1 “Biochemical composition” - there you will find the initial content of proteins, fat, sugars in the investigated CW.
- The conclusions have been substantially reworded in response to the suggestions of all the Reviewers (there were different suggestions).
Best regards,
Agnieszka Pilarska

Round 2
Reviewer 1 Report
The authors have improved the quality of the manuscript after including a statistical analysis of BMP test data and finding a mathematical correlation between protein content and BMP data. Also, some recent literature references have been added. Therefore, my recommendation is to accept this work for publication.
Author Response
Dear Reviewer,
thank you very much for your recommendation.
Best regards,
Agnieszka Pilarska

Reviewer 3 Report
The manuscript has been improved, although there are still minor English amendments that need to be done, For example, authors should not use "his" or "her" when referring to another work. Also, it is better to use "present" instead of "presented study" . E.g. "present study".
In general, the MS can be now published.
Author Response
Dear Reviewer,
thank you very much for your recommendation. I corrected my article according your suggestions.
The changes were marked in green.
Best regards,
Agnieszka Pilarska
